# Scale Contrastive Learning with Selective Attentions for Blind Image Quality Assessment

## Abstract

Human visual perception naturally evaluates image quality across multiple scales, a hierarchical process that existing blind image quality assessment (BIQA) algorithms struggle to replicate effectively. This limitation stems from a fundamental misunderstanding: current multi-scale approaches fail to recognize that quality perception varies dramatically between scales—what appears degraded when viewed closely may look acceptable from a distance. This inconsistency not only creates misleading "visual illusions" during feature fusion but also introduces substantial redundant information that dilutes quality-critical features and leads to imprecise assessments. Our CSFIQA framework advances multi-scale BIQA via two key innovations: (1) a selective focus attention mechanism that mimics human visual attention by filtering out redundant cross-scale information that would otherwise mask subtle quality indicators, and (2) a scale contrastive learning strategy that explicitly learns to distinguish quality variations both across and within scales. By incorporating an adaptive noise sample matching mechanism, CSFIQA effectively identifies perceptual quality discrepancies in the same content viewed at different scales. Experiments demonstrate substantial improvements over state-of-the-art methods across seven datasets, achieving up to 8.8% SRCC improvement on challenging real-world distortions, confirming CSFIQA's superior alignment with human quality perception.

## 1 Introduction

Image Quality Assessment (IQA) aims to model the human visual system's ability to perceive image quality Yang et al. (2022); Zhang et al. (2022b). It has been widely applied in fields such as image restoration Zhang et al. (2022a), compression Liu et al. (2022), and generation Wang et al. (2023b), with the goal of enhancing human visual experience. Based on whether distortion-free reference images are required, IQA can be classified into three types: full-reference, reduced-reference, and no-reference (or blind) IQA Liu et al. (2024); Chahine et al. (2023); Zhang et al. (2023). Among these, Blind Image Quality Assessment (BIQA) methods have received increasing attention due to their broad applicability.

However, BIQA faces a fundamental yet largely overlooked challenge: the dramatic variation in quality perception across different scales of the same image. While humans effortlessly integrate these multi-scale impressions Chen et al. (2024), current BIQA algorithms struggle with this inherent complexity, creating a significant gap between algorithmic assessments and human judgment. This perceptual discrepancy severely limits the effectiveness of BIQA in critical applications ranging from image compression and restoration to content generation systems. Current multi-scale BIQA methods, whether operating at the image level through direct resizing (Fig. 2(b)) or at the feature level through pyramid structures (Fig. 2(c)), encounter two critical bottlenecks that fundamentally undermine their performance:

First, the "visual illusions" problem presents a severe limitation in existing approaches. Traditional methods erroneously assume that image regions maintain consistent quality characteristics regardless of scale—a fundamentally flawed assumption illustrated in Fig. 2(a). In reality, quality perception varies dramatically across scales: compression artifacts that dominate perception at a large scale

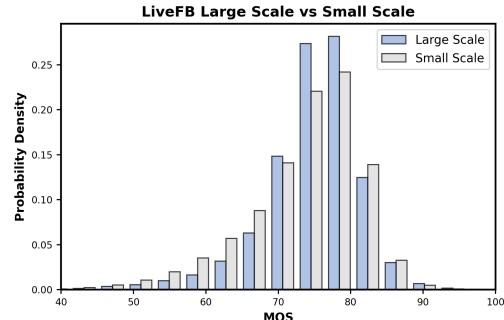

(a) Visualizations of images at different scales that correspond to different qualities (b) Image Pyramid BIQA (c) Feature Pyramid BIQA (d) Our Feature Pyramid Approach

Figure 2: (a) The same image under different patch perspectives can lead to varying quality judgments, and simply combining information from different viewpoints is prone to causing visual hallucinations. (b-d) Comparison of mainstream multi-scale paradigms with our approach, which uses scale contrastive learning to distinguish quality differences in (a). The designed selective focus attention can remove redundant semantic information and enhance attention related to perceptual quality.

(close-up view) may become imperceptible at smaller scales (distant view), while global distortions like overexposure might dominate at smaller scales but remain unnoticed in large-scale patches. When algorithms naively combine features from multiple scales, they generate misleading quality representations where high-quality and low-quality signals from different scales incorrectly neutralize each other. This creates a perceptual distortion we term the "visual illusions" effect—where the algorithm perceives uniform quality across an image despite significant scale-dependent variations—leading to assessments that fundamentally contradict human perception.

To support this claim, Fig.1 presents the Mean Opinion Score (MOS) distributions of the LiveFB dataset at different scales, clearly demonstrating the existence of scale-sensitive perceptual differences. Building upon this observation, Tab.7 further shows that neglecting such scale-specific variations during training—by assigning uniform quality labels across scales—can degrade model performance, thereby highlighting the necessity of incorporating scale-aware modeling in IQA. In addition, our analysis of feature activation maps (Fig. 6) provides a more intuitive visualization of how this illusion effect manifests in practice: current models tend to focus on undistorted regions while neglecting areas that are more indicative of actual image quality degradation.

Second, the "information dilution" problem significantly impairs quality detection sensitivity. Unlike human vision, which selectively focuses on quality-relevant features while filtering redundant information, current multi-scale methods indiscriminately process all cross-scale information. This approach not only wastes computational resources but, more critically, creates a signal-to-noise problem where quality-critical features become overwhelmed by redundant semantic content shared across scales. As demonstrated in our visualization results (Fig. 6), this dilution effect causes quality indicators—particularly subtle artifacts and distortions—to be masked by dominant semantic features that remain consistent across scales but contribute little to quality assessment. This effect is especially pronounced in authentic distortion datasets where quality variations are complex and multifaceted.

To address these fundamental limitations, we propose CSFIQA (Contrast-Constrained Scale-Focused Image Quality Assessment), a novel framework specifically designed to accurately model scale-dependent quality perception. Our approach introduces two complementary innovations:

The first innovation directly tackles the "information dilution" problem through a Selective Focus Attention (SFA) mechanism. This mechanism intelligently identifies and filters redundant cross-scale information by preserving only the most relevant attention values through an adaptive filtering selector. It then amplifies quality-discriminative features through an information concentrator module, effectively mimicking the human visual system's ability to focus attention on perceptually important regions while suppress-

Figure 1: The MOS distribution of LiveFB in large scale (40% of the original image size) and small scale (20% of the original image size).

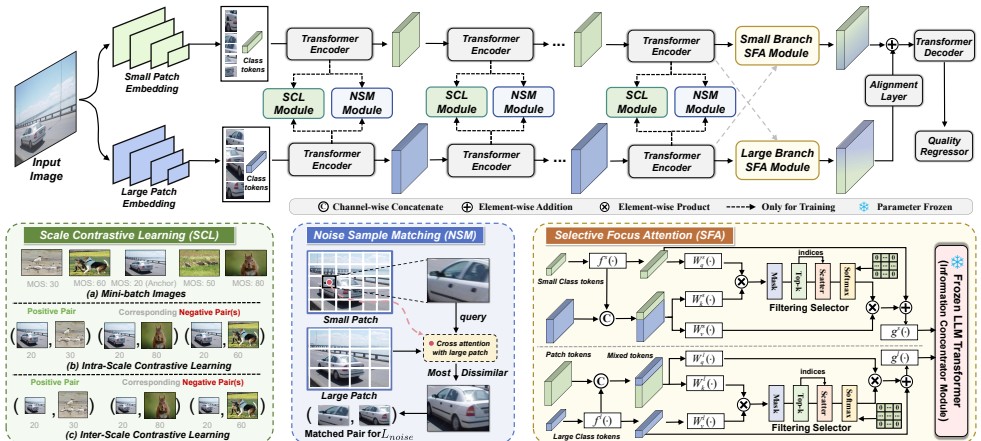

Figure 3: In our CSFIQA framework, a Transformer Encoder extracts multi-scale image features ($F_a^l$). These are input to the SCL module (Sec. 3.2), which uses inter- and intra-scale contrastive learning to enhance quality discrimination. The NSM module (Sec. 3.2) mitigates the "visual illusions" effect (see Fig. 2(a)) by distinguishing subtle regional quality variations. Subsequently, features from the final layer ($F_a^L$) enter the SFA module (Sec. 3.3), where an Adaptive Filtering Selector (AFS) and Information Concentrator Module (ICM) create quality-aware features. Finally, a decoder predicts the quality score $\hat{Y}$ (see Algorithm 1 for details).

ing irrelevant information. This approach significantly enhances the model's ability to isolate and emphasize quality-critical features that would otherwise be overwhelmed by redundant semantic content.

The second innovation addresses the "visual illusions" problem through a comprehensive Scale Contrastive Learning (SCL) framework with an adaptive Noise Sample Matching (NSM) mechanism. This approach explicitly teaches the model to distinguish between different quality characteristics at both inter-scale (across different scales) and intra-scale (within the same scale) levels. Crucially, the NSM mechanism specifically targets regions with inconsistent quality information across scales within the same image, enabling the model to accurately represent scale-dependent quality variations rather than incorrectly averaging them. This explicit modeling of scale-quality relationships effectively prevents the visual illusion effect that plagues traditional multi-scale approaches.

Our comprehensive evaluation on seven benchmark datasets demonstrates that CSFIQA consistently outperforms state-of-the-art methods, with particularly significant improvements on challenging real-world datasets: 8.8% SRCC improvement on LIVEFB and 1.6% on LIVEC. These results confirm that by accurately addressing the fundamental limitations of scale-dependent quality perception, our approach successfully bridges the gap between algorithmic assessment and human perception of image quality.

## 2 RELATED WORK

### 2.1 MULTI-SCALE AND CONTRASTIVE LEARNING IN BIQA

Deep learning has significantly advanced Blind Image Quality Assessment (BIQA), with models evolving from Convolutional Neural Networks (CNNs) to Vision Transformers (ViTs). While CNNs excel at extracting local features (Bosse et al., 2017), they struggle to capture long-range dependencies. ViTs and hybrid architectures (Qin et al., 2023; Golestaneh et al., 2022b) overcome this using self-attention but can introduce parameter redundancy or over-emphasize global semantics at the expense of sensitivity to local distortions. To address these limitations, multi-scale architectures have become central to BIQA, mimicking the human ability to integrate fine details with global context (Su et al., 2020b; Chen et al., 2024). Despite their progress, these methods often fail to effectively model complex, scale-sensitive quality discrepancies and can introduce redundant in-

formation, highlighting the need for a more refined mechanism to balance local fidelity and global context.

Alongside architectural evolution, contrastive learning has emerged as a powerful paradigm for learning robust quality representations (Zhao et al., 2023; Li et al., 2024). Seminal works like CON-TRIQUE (Madhusudana et al., 2022) and Re-IQA (Saha et al., 2023) demonstrated its effectiveness. However, these methods typically rely on distortion types or levels as supervisory signals, rather than ground-truth quality scores. They also tend to treat any two images with different content as a negative pair, overlooking the nuanced quality relationships that might exist between them. Our work addresses these gaps by introducing scale contrastive learning, which directly models the relationship between quality at different resolutions and the overall perceptual score.el the relationship between different quality regions and the overall image quality.

## 3 PROPOSED METHOD

### 3.1 OVERALL PIPELINE

We proposed CSFIQA, which is designed to precisely model the relationship between image scale and overall quality. As illustrated in Fig. 3, CSFIQA integrates three primary modules: the Scale Contrastive Learning module **(SCL)**, the Noise Sample Matching module **(NSM)**, and the Selective Focus Attention module **(SFA)**. Initially, the input image $I$ is segmented into patches of varying scales, which are independently processed through distinct $L$ layers of the Transformer encoder to extract scale-level features $F_a^l$ ($a \in \{small, large\}$) at the $l$-th layer. The scale-level features are subsequently forwarded to the SCL (Sec. 3.2) to derive positive and negative samples, denoted as $P$ and $N$. They are utilized to compute their respective InfoNCE loss. Simultaneously, through NSM module (Sec. 3.2), we calculate the similarity between image patches with the largest quality gap within the same image as a loss to distinguish them. The output of the final encoder layer is passed to the SFA (Sec. 3.3) to filter redundant features and amplify quality features. Finally, we obtain multi-scale quality features, refined via an alignment layer and Transformer decoder to yield predicted quality scores (see Algorithm 1).

### 3.2 SCALE CONTRASTIVE LEARNING

To enhance the model's perception of scale-dependent quality, we introduce Intra- and Inter-Scale contrastive learning. This approach ensures that representations for images of similar quality are consistent across different scales, while representations for images of dissimilar quality are pushed apart. Consequently, the model can better integrate fine-scale features, which are sensitive to local distortions like noise and artifacts, with coarse-scale features that capture global distortions like overexposure and blur.

As shown in the SCL module (Fig. 3c), we use Mean Opinion Score (MOS) similarity to select positive and negative sample pairs, aligning feature representations with perceptual quality. Specifically, for a given query patch $i$ and another patch $j$ in a mini-batch of size $B$, we start with their MOS vector $y \in \mathbb{R}^B$. We then compute a pairwise score distance matrix $Y_d \in \mathbb{R}^{B \times B}$ using the Manhattan distance. Let $Y_d^{ij}$ be the distance between the scores of $i$ and $j$, and let $Y_{d,\max}^i = \max_k Y_d^{ik}$ be the maximum distance in the query's row. Using two threshold coefficients, $\gamma_1$ and $\gamma_2$, we define the sample classifier as:

$$Classifier(i,j) \in \begin{cases} P, & if \ Y_d^{ij} \le \gamma_1 * Y_d^{\hat{i}}, \\ N, & if \ Y_d^{ij} > (1 - \gamma_2) * Y_d^{\hat{i}}. \end{cases} \tag{1}$$

Therefore, given a pacth with feature $F_a^l \in P$, we define scale-level contrastive loss as follows:

$$\mathcal{L}_{scale} = \sum_{l=1}^{L} \sum_{a \in \{s,l\}} \frac{1}{|P|} \sum_{F^+ \in P} \mathcal{L}(F_a^l, F^+), \tag{2}$$

where

$$\mathcal{L}(F_a^l, F^+) = \log \frac{- \exp(\frac{F_a^l \cdot F^+}{\tau})}{\exp(\frac{F_a^l \cdot F^+}{\tau}) + \sum_{F^- \in N} \exp(\frac{F_a^l \cdot F^-}{\tau})}. \tag{3}$$

---

**Algorithm 1** Pseudocode for proposed CSFIQA

---

1: **Input**: Mini-batch Images $I = \{X_i, Y_i\}_{i=1}^B$;
2: **Variables:** $B$: Batch size; $F_a^i$: Image feature; $P$: Positive pairs; $N$: Negative pairs; $M$: Small patch regions by NSM; $K^{'}$: Neighbouring large patch regions.
3: **Output**: Predicted quality score $\{\hat{Y}\}_{i=1}^B$
4: **for** `l, blk in enumerate(blocks)` **do**
5:     *// Transformer encoder*
6:     **for** $i = 1$ to $B$ **do**
7:       *// Calculate the P and N pairs by Eq.3*
8:       $\mathcal{L}(F_a^l, F^+) = \log \frac{-\exp(\frac{F_a^l \cdot F^+}{\tau})}{\exp(\frac{F_a^l \cdot F^+}{\tau}) + \sum_{F^- \in N} \exp(\frac{F_a^l \cdot F^-}{\tau})}$
9:       $\mathcal{L}_{scale} += \mathcal{L}(F_a^l, F^+)$
10:       **for** $m = 1$ to $M$ **do**
11:         $K^{'} = \text{neighbouring}(m)$
12:         **for** $k = 1$ to $K^{'}$ **do**
13:           $Sim(G_{small}^m, G_{large}^k) = \frac{G_{small}^m \cdot G_{large}^k}{\|G_{small}^m\|\|G_{large}^k\|}$
14:           $\mathcal{L}_{noise} += \frac{1}{\exp(Sim(G_{small}^m, G_{large}^k)}$
15:         **end for**
16:       **end for**
17:     **end for**
18:     *// Selective Focus Attention*
19:     $F_a = SFA(F_a^B)$
20: **end for**
21: *// Obtain F from $F_a$ through the Alignment Layer*
22: $\hat{Y} = Decoder(F)$ *// Transformer decoder*
23: $\mathcal{L}' = \left( \left\| \hat{Y} - Y \right\|_1 + \lambda \left( \mathcal{L}_{scale} + \mathcal{L}_{noise} \right) \right)$

---

Here, $\tau$ denotes the temperature hyperparameter and $F_a^l$ represents the features of patch at any scale in layer $l$ of the encoder. Empirically, we set $\gamma_1$ to 0.2, $\gamma_2$ to 0.7, and $\tau$ to 0.3. Ablation studies on hyperparameter settings are presented in Tab. 6 and Tab. 5. Analysis and discussion of computational costs can be found in the **Appendix.**

**Noise Sample Matching (NSM).** We propose a simple but effective adaptive noise sample matching mechanism to further distinguish samples with inconsistent quality information across different scales of the same image. We identify the sample in the neighbouring region at scale $s$ with the least similar quality information as a negative sample for contrastive learning for a given image at scale $l$.

First, we divide the feature maps into regions for different scales. We obtain features with $G_a \in \mathbb{R}^{N_a \times D_a}$ from the patch embedding in the ViT encoder and reorganize these patches into a feature map $\hat{G}_a \in \mathbb{R}^{H_a \times W_a \times D_a}$ with equal height and width. We then apply a sliding window function $W_a \in \mathbb{R}^{H_a' \times W_a'}$ to further partition these patches into regions. Simultaneously, based on spatial coordinates, we record the large patch regions that encompass each small pacth region, designating them as neighboring regions corresponding to each small patch region. We define $M$ blocks obtained by partitioning at the $small$ patch and $K$ blocks obtained by partitioning at the $large$ patch, wherein the number of neighboring large patch regions for each small patch is $K^{'} (K^{'} \leq K)$. For each region $G_{small}^m$ at $small$ patch, we compute its cosine similarity with each neighbouring region $G_{large}^k$ at $large$ patch:

$$Sim(G_{small}^m, G_{large}^k) = \frac{G_{small}^m \cdot G_{large}^k}{\|G_{small}^m\|\|G_{large}^k\|}. \tag{4}$$

We compute the loss across all neighboring regions in the small patch feature map, as shown in Eq. (5). This approach enables us to amplify the distance between samples with inconsistent quality information at different scales within the same image, thereby more effectively differentiating between them.

$$\mathcal{L}_{noise} = \sum_{m=1}^M \sum_{k=1}^{K^{'}} \frac{1}{\exp(Sim(G_{small}^m, G_{large}^k))}. \tag{5}$$

**Overall Loss.** Let $\hat{Y}$ and $Y$ respectively denote the predicted scores and the ground truth scores for the image $I$. Given $\lambda$ represent the hyperparameters. The notation $\|\cdot\|_1$ signifies the $\ell_1$ regression loss. The total loss is defined as:

$$\mathcal{L} = \sum_I \left( \left\| \hat{Y} - Y \right\|_1 + \lambda \left( \mathcal{L}_{scale} + \mathcal{L}_{noise} \right) \right). \tag{6}$$

### 3.3 Selective Focus Attention

**Preliminaries.** Traditional multi-scale Image Quality Assessment (IQA) methods often suffer from information redundancy, causing them to overlook critical quality-related features. To address this, we propose the **Selective Focus Attention (SFA)** module. The SFA consists of two sequential components: an **Adaptive Filtering Selector (AFS)** and an **Information Concentrator Module (ICM)**. The AFS first employs a filtering attention mechanism to select the most salient cross-scale information. Subsequently, the ICM, which combines a learnable linear layer with a frozen large language model, refines these selected features to pinpoint quality-specific content, thus reducing redundancy in the process. We begin by examining the cross-attention mechanism commonly used in multi-scale models. For different branches labeled as large and small, the class token from the large branch is concatenated with the patch token from the small branch:

$$x' = \left[ x_{\text{cls}}^{large}, x_{\text{patch}}^{small} \right],$$

$$CrossAtt(x') = \text{softmax} \left( \frac{QK^T}{\sqrt{C/h}} \right) V. \tag{7}$$

Here, $C$ represents the number of channels, and $h$ denotes the number of heads. Given $Q = x_{\text{cls}}' W_q$, $K = x' W_k$, and $V = x' W_v$, this enables the fusion of cross-scale features.

**Adaptive Filtering Selector (AFS).** The core of our approach is the AFS mechanism, which enhances the standard attention computation from Eq. 7. Instead of using the full attention matrix, AFS implements an **adaptive top-k filtering** strategy by applying a learnable masking operator, $\mathcal{M}$, to the raw attention scores. For each query, this operator dynamically selects the top-$k$ most relevant key-value pairs. The value of $k$ is not fixed but is determined by a learnable parameter constrained to a fractional range $[\alpha, \beta]$ of the total tokens. This allows the model to adaptively decide how much information to prune. Attention scores not within the top-$k$ are masked with $-\infty$ before the softmax function, effectively nullifying their contribution. The filtering mechanism is formally defined as:

$$SelectAtt(Q, K, V) = \text{softmax} \left( \mathcal{M} \left( \frac{QK^\top}{\sqrt{d}} \right) \right) V. \tag{8}$$

Here, $\mathcal{M}$ is the top-k operator, with $[\alpha, \beta] = [1/3, 3/4]$.

**Information Concentrator Module (ICM).** Recent work (Pang et al., 2023) has shown that frozen LLM encoders can discern information-rich visual tokens and further enhance their contributions to latent representations. In our approach, we have implemented the filtering of features at the scale level before inputting them into the frozen LLM layer. As a result, the frozen LLM layer functions as a scale information amplifier, exhibiting a stronger focus on the feature content that we consider essential. The specific structure of the Information Concentrator and the visualization results of the SFA will be presented in the **Appendix**.

## 4 Experiments

### 4.1 Datasets and Evaluation Protocols

We evaluated our model on eight public Image Quality Assessment (IQA) datasets. Four datasets feature authentic distortions: LIVEC (Ghadiyaram & Bovik, 2015), KonIQ-10k (Hosu et al., 2020), LIVEFB (Ying et al., 2020), and SPAQ (Fang et al., 2020). The other four contain synthetic distortions: LIVE (Sheikh et al., 2006), CSIQ (Larson & Chandler, 2010), TID2013 (Ponomarenko et al., 2015), and KADID (Lin et al., 2019). These datasets vary significantly in scale and content, from hundreds of images with a few distortion types to nearly 40,000 images with diverse artifacts. To

Table 1: Performance comparison based on average SRCC and PLCC. Bold values denote the best and second-best results.

| Method | LIVE | | CSIQ | | TID2013 | | LIVEC | | KonIQ | | LIVEFB | | SPAQ | |
|---|---|---|---|---|---|---|---|---|---|---|---|---|---|---|
| | PLCC | SRCC | PLCC | SRCC | PLCC | SRCC | PLCC | SRCC | PLCC | SRCC | PLCC | SRCC | PLCC | SRCC |
| BRISQUE (Mittal et al., 2012) | 0.944 | 0.929 | 0.748 | 0.812 | 0.571 | 0.626 | 0.629 | 0.629 | 0.685 | 0.681 | 0.341 | 0.303 | 0.817 | 0.809 |
| ILNIQE (Zhang et al., 2015) | 0.906 | 0.902 | 0.865 | 0.822 | 0.648 | 0.521 | 0.508 | 0.508 | 0.537 | 0.523 | 0.332 | 0.294 | 0.712 | 0.713 |
| BIECON (Kim & Lee, 2016) | 0.961 | 0.958 | 0.823 | 0.815 | 0.762 | 0.717 | 0.613 | 0.613 | 0.654 | 0.651 | 0.428 | 0.407 | - | - |
| MEON (Ma et al., 2017) | 0.955 | 0.951 | 0.864 | 0.852 | 0.824 | 0.808 | 0.710 | 0.697 | 0.628 | 0.611 | 0.394 | 0.365 | - | - |
| DBCNN (Zhang et al., 2018) | 0.971 | 0.968 | 0.959 | 0.946 | 0.865 | 0.816 | 0.869 | 0.851 | 0.884 | 0.875 | 0.551 | 0.545 | 0.915 | 0.911 |
| MetaIQA (Zhu et al., 2020) | 0.959 | 0.960 | 0.908 | 0.899 | 0.868 | 0.856 | 0.802 | 0.835 | 0.856 | 0.887 | 0.507 | 0.54 | - | - |
| P2P-BM (Ying et al., 2020) | 0.958 | 0.959 | 0.902 | 0.899 | 0.856 | 0.862 | 0.842 | 0.844 | 0.885 | 0.872 | 0.598 | 0.526 | - | - |
| HyperIQA (Su et al., 2020a) | 0.966 | 0.962 | 0.942 | 0.923 | 0.858 | 0.840 | 0.882 | 0.859 | 0.917 | 0.906 | 0.602 | 0.544 | 0.915 | 0.911 |
| MUSIQ (Ke et al., 2021) | 0.911 | 0.940 | 0.893 | 0.871 | 0.815 | 0.773 | 0.828 | 0.785 | 0.928 | 0.916 | 0.661 | 0.566 | 0.921 | 0.918 |
| TReS (Golestaneh et al., 2022a) | 0.968 | 0.969 | 0.942 | 0.922 | 0.883 | 0.863 | 0.882 | 0.859 | 0.928 | 0.915 | 0.625 | 0.554 | - | - |
| DACNN (Pan et al., 2022) | 0.980 | 0.978 | 0.957 | 0.943 | 0.889 | 0.871 | 0.884 | 0.866 | 0.912 | 0.901 | - | - | 0.921 | 0.915 |
| Re-IQA (Saha et al., 2023) | 0.971 | 0.970 | 0.96 | 0.947 | 0.861 | 0.804 | 0.854 | 0.84 | 0.923 | 0.914 | - | - | 0.925 | 0.918 |
| DEIQT (Qin et al., 2023) | **0.982** | 0.980 | 0.963 | 0.946 | **0.908** | 0.892 | 0.894 | 0.875 | 0.934 | 0.921 | 0.663 | 0.571 | 0.923 | 0.919 |
| CLIP-IQA+ (Wang et al., 2023a) | - | - | - | - | - | - | 0.832 | 0.805 | 0.909 | 0.895 | 0.593 | 0.575 | 0.866 | 0.864 |
| CDINet (Zheng et al., 2024) | 0.975 | 0.977 | 0.960 | 0.952 | **0.908** | **0.898** | 0.880 | 0.865 | 0.928 | 0.916 | - | - | 0.922 | 0.919 |
| QFM-IQM (Li et al., 2025) | **0.983** | **0.981** | **0.965** | **0.954** | - | - | **0.913** | **0.891** | 0.936 | 0.922 | 0.667 | 0.567 | 0.924 | **0.920** |
| LoDa (Xu et al., 2024) | 0.979 | 0.975 | - | - | 0.901 | 0.869 | 0.899 | 0.876 | **0.944** | **0.932** | **0.679** | **0.578** | 0.928 | 0.925 |
| CSFIQA (ours) | **0.983** | **0.982** | **0.973** | **0.967** | **0.917** | **0.899** | **0.922** | **0.905** | **0.944** | **0.924** | **0.701** | **0.629** | **0.935** | **0.925** |

Table 2: SRCC on the cross datasets validation. The best performances are highlighted in bold.

| Training | LIVEFB | | LIVEC | KonIQ | LIVE | CSIQ |
|---|---|---|---|---|---|---|
| Testing | KonIQ | LIVEC | KonIQ | LIVEC | CSIQ | LIVE |
| DBCNN | 0.716 | 0.724 | 0.754 | 0.755 | 0.758 | 0.877 |
| P2P-BM | 0.755 | 0.738 | 0.740 | 0.770 | 0.712 | - |
| TReS | 0.713 | 0.74 | 0.733 | 0.786 | 0.761 | - |
| DEIQT | 0.733 | 0.781 | 0.744 | 0.794 | 0.781 | 0.932 |
| LoDa | 0.763 | **0.805** | 0.745 | 0.811 | - | - |
| ours | **0.785** | **0.805** | **0.762** | **0.838** | **0.786** | **0.933** |

Table 3: Ablation experiments on different modules.

| Method | | | LIVEC | | CSIQ | |
|---|---|---|---|---|---|---|
| Baseline | SCL | SFA | PLCC | SRCC | PLCC | SRCC |
| ✔ | | | 0.896 | 0.878 | 0.964 | 0.948 |
| ✔ | ✔ | | **0.911** | **0.892** | **0.970** | **0.963** |
| ✔ | | ✔ | 0.904 | 0.887 | 0.965 | 0.956 |
| ✔ | ✔ | ✔ | 0.922 | 0.905 | 0.973 | 0.967 |

measure performance, we used the Spearman Rank Correlation Coefficient (SRCC) and the Pearson Linear Correlation Coefficient (PLCC). For both metrics, values closer to 1 indicate superior predictive performance.

## 4.2 IMPLEMENTATION DETAILS AND SETUPS

We use a pre-trained CrossViT (Chen et al., 2021) as our scale encoder. The small and large-scale branches have depths of 1 and 4, respectively, using patch sizes of 12 and 16, and token dimensions of 192 and 384. The number of attention heads is 6. For our frozen LLM module, we employ Llama-7B. We also include a standard transformer decoder with a depth of 1 for baseline comparisons. The model was trained for 9 epochs with a learning rate 2e-4 and a decay factor of 10 applied every 3 epochs. Batch sizes range from 16 to 128, depending on the dataset size. The dataset was divided into 80% for training and 20% for testing, and this process was repeated ten times to minimize performance bias. For other hyperparameters, we set $\lambda$ to 0.01, $[\alpha, \beta]$ to [1/3, 3/4], $\gamma_1$ to 0.2, $\gamma_2$ to 0.7, and $\tau$ to 0.3. We report the median SRCC and PLCC to evaluate the model's prediction accuracy and monotonicity performance. All experiments were conducted on eight NVIDIA RTX 3090 GPUs.

## 4.3 PERFORMANCE COMPARISON WITH SOTA

As shown in Tab. 1, we compare our CSFIQA against 16 classical and state-of-the-art BIQA methods. For competing models, we used publicly available implementations or retrained them with official code. The comparison includes both hand-crafted feature methods (e.g., BRISQUE, ILNIQE) and deep learning approaches (e.g., LoDa, QFM-IQM). CSFIQA outperforms all competing methods on six of the seven datasets, demonstrating robust performance across diverse image content and distortion types. The improvement is particularly stark on the challenging LIVEFB dataset, where our model achieves an 8.8% performance gain. We attribute this success to our model's superior ability to distinguish quality information across different scales, a challenge prominent in the LIVEFB dataset. These results confirm the effectiveness and superiority of our method.

Table 4: Ablation experiments with different components on two datasets.

| Module | w/ Sub-Modules | LIVEC | | CSIQ | |
|---|---|---|---|---|---|
| | | PLCC | SRCC | PLCC | SRCC |
| w/ SCL | inter-SCL | 0.899 | 0.883 | 0.965 | 0.951 |
| | intra-SCL | 0.902 | 0.884 | 0.965 | 0.954 |
| | **NSM** | **0.903** | **0.887** | **0.967** | **0.958** |
| w/ SFA | **AFS** | **0.902** | **0.884** | **0.967** | **0.953** |
| | ICM | 0.899 | 0.882 | 0.965 | 0.950 |

Table 5: Ablation study about $\lambda$ in Eq. 6.

| hyperparameter $\lambda$ | LIVEC | | KonIQ | |
|---|---|---|---|---|
| | PLCC | SRCC | PLCC | SRCC |
| 1 | 0.908 | 0.894 | 0.911 | 0.932 |
| 0.1 | 0.915 | 0.900 | 0.917 | 0.940 |
| 0.01 | **0.922** | **0.905** | **0.924** | **0.944** |
| 0.001 | 0.910 | 0.887 | 0.922 | 0.941 |
| 0.0001 | 0.909 | 0.893 | 0.918 | 0.938 |

## 4.4 GENERALIZATION CAPABILITY VALIDATION

To validate the generalization of our model, CS-FIQA, we conducted cross-dataset experiments, training on one dataset and testing on others without any fine-tuning. As shown in Tab. 2, which reports the average SRCC scores, CSFIQA achieved the best performance in all evaluations, with particularly strong results on the LIVEC and KonIQ datasets. We attribute this robust generalization to our model's unique architecture. The SCL and NSM modules excel at differentiating quality variations across scales, while the SFA module effectively focuses on the most salient information. This confirms our model's exceptional

Table 6: Ablation about the $[\alpha, \beta]$ in AFS.

| range $[\alpha, \beta]$ | LIVEC | | KonIQ | |
|---|---|---|---|---|
| | PLCC | SRCC | PLCC | SRCC |
| $[1/2]$ | 0.899 | 0.872 | 0.912 | 0.896 |
| $[1/6, 1/3]$ | 0.903 | 0.885 | 0.928 | 0.902 |
| $[1/5, 1/2]$ | 0.906 | 0.891 | 0.935 | 0.915 |
| $[1/4, 2/3]$ | 0.916 | 0.898 | 0.941 | 0.919 |
| $[1/3, 3/4]$ | **0.922** | **0.905** | **0.944** | **0.924** |
| $[1/2, 4/5]$ | 0.918 | 0.899 | 0.938 | 0.916 |
| $[2/3, 1]$ | 0.908 | 0.887 | 0.927 | 0.902 |

generalization ability. For further validation, results from cross-distortion tests are available in the **Appendix**.

## 4.5 ABLATION STUDY

**Overall.** Tab. 3 and Tab. 4 present the ablation performance of our proposed main framework, Scale Contrastive Learning (SCL) and Selective Focus Attention (SFA), along with their sub-modules on the LIVEC dataset. Our SCL framework primarily consists of Intra/Inter Scale Learning modules and a Noise Sample Matching (NSM) mechanism. Selective Focus Attention primarily consists of the Adaptive Filtering Selector (AFS) module for redundant information filtering and the Information Concentrator Module (ICM) for amplifying quality-relevant information. In Tab 3, we employ CrossViT (supplemented with a transformer decoder) as our baseline model. Tab 4 further refines the contribution of each sub-module to performance improvement. Tab. 5 and Tab. 6 present the performance of our hyperparameters on LIVEC and KonIQ datasets. Notably, except for the observed hyperparameters, all other hyperparameters were selected according to the settings reported in Sec. 4.2.

**Effect of Scale Contrastive learning Module (SCL).** Tab. 3 demonstrates that SCL provides the most significant improvement to our method. This further illustrates SCL's effectiveness in mitigating the "visual illusion" problem. Specifically, inter/intra-SCL helps the model establish quality relationships between different images from varied perspectives, strengthening its quality perception capabilities across arbitrary scales. The NSM module is explicitly designed to address the "visual illusion" problem by bringing together features from different regions to distinguish subtle quality variations within the same image at different scales. The substantial improvement shown by the NSM module in Tab. 4 confirms this effectiveness.

**Effect of Selective Focus Attention (SFA).** Tab. 3 demonstrates how the SFA module enhances quality assessment by removing redundant semantic information. This is further evidenced in Tab. 4, where the AFS module shows greater performance gains compared to the ICM module. This further demonstrates that the performance gain from focusing becomes evident once redundant information is eliminated.

**Effect of weight $\lambda$.** We use $\lambda$ in Eq. 6 to balance the scale contrastive learning. To this end, we conducted a sensitivity analysis on different values of $\lambda$ to investigate the effect of inter-scale contrastive learning. As shown in Tab. 5, we found that smaller values weaken inter-scale contrastive

learning, while larger values cause excessive feature space changes and performance degradation, resulting in performance degradation. Therefore, we ultimately set $\lambda$=0.01.

**Effect of weight** $[\alpha, \beta]$**.** Tab. 6 presents our ablation study on the AFS module's filtering range, $[\alpha, \beta]$. The choice of $k$ is critical for performance; we found that using a single, fixed value for $k$ resulted in instability. To enhance robustness, we instead sample $k$ from a learnable range $[\alpha, \beta]$. The results reveal a clear trade-off. A range set too low results in insufficient information aggregation, causing a sharp

Table 7: Impact of ignoring scale-sensitive MOS labels during training.

| Method | Diff. Scale | Diff. MOS | PLCC | SRCC |
|--------|:-----------:|:---------:|:-----:|:-----:|
| **CSFIQA** | ✔ | ✗ | 0.710 | 0.641 |
| **CSFIQA** | ✔ | ✔ | 0.733 | 0.691 |

performance decline. Conversely, a range set too high introduces semantically irrelevant information, which also degrades performance. We achieved optimal results with $[\alpha, \beta]$ set to **[1/3, 3/4]**, confirming this range provides the best balance of focused yet sufficient information.

### 4.6 QUALITATIVE ANALYSIS

**Feature Visualization.** Fig. 6 presents GradCAM (Selvaraju et al., 2017) visualizations comparing the feature attention of our model, CSFIQA, against the baseline. The results clearly show that CSFIQA accurately focuses on distorted image regions and accurately predict the quality score.

In contrast, the baseline is often distracted by irrelevant content, a phenomenon we term "visual illusion", which impairs its judgment. This improved focus stems from our model's ability to effectively process and integrate quality information from different scales, an ability deliberately cultivated during training.

Notably, the last two rows display our model's visualization at each distinct scale. In these "visual illusion" cases, the perceived quality regions for the same image vary significantly across scales, leading to a large discrepancy in their quality assessments. This phenomenon aligns perfectly with our research motivation. Enlarged visualizations are in the **Appendix.**

**Scale Quantitative Analysis.** To investigate the impact of scale variations on image quality assessment, we conducted experiments using two settings in the CSFIQA framework, as shown in Tab. 7. The first setting, Different Scale / Same MOS, assigns the same quality label across varying scales, while the second setting, Different Scale / Different MOS, assigns distinct quality labels

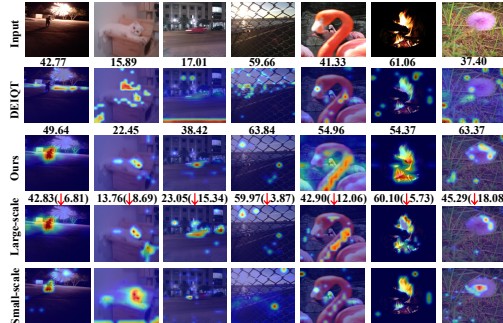

Figure 4: Grad-CAM Activation Maps of DEIQT and CSFIQA on LIVEC dataset. Scores below the first row indicate ground-truth MOS. Our model focuses more on distorted regions, leading to predictions closer to true values. Rows 1–3: input image, baseline CAM, CSFIQA CAM. Rows 4–5: large and small scale feature visualizations of CSFIQA.

based on scale differences. The results, evaluated on the LiveFB dataset, show that the Different Scale / Different MOS setting outperforms Different Scale / Same MOS, emphasizing the importance of incorporating scale-aware features in quality prediction.

### 5 CONCLUSION

In this study, we introduce the Contrast-Constrained Scale-Focused IQA Framework (CSFIQA), designed to capture quality information across diverse regions of an image effectively. Unlike traditional models that merely concatenate scale information, CSFIQA leverages cross-scale contrastive learning to differentiate the varying quality within a single image. Additionally, we implement a selective focus attention mechanism to refine quality information. Experiments show that CSFIQA surpasses existing BIQA methods.

## ETHICS STATEMENT

This work adheres to the ICLR Code of Ethics. In this study, no human subjects or animal experimentation was involved. All datasets used were sourced in compliance with relevant usage guidelines, ensuring no violation of privacy. We have taken care not to achieve any bias or discriminatory outcomes in our research process. No personally identifiable information was used, and no experiments were conducted that could raise privacy or security concerns. We are committed to maintaining transparency and integrity throughout the research process.

## REPRODUCIBILITY STATEMENT

To ensure the reproducibility of this study, we provide the source code of the proposed model along with the training and evaluation scripts in the supplementary materials. The implementation details, hyper-parameters, and experimental settings described in Sec. 4.2 of the main paper are sufficient to reproduce the reported results. In addition, all IQA benchmark datasets are publicly available, ensuring consistent and reproducible evaluation outcomes. We believe these measures will enable other researchers to replicate our work and further advance the field.

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

# A APPENDIX

## A.1 THE USE OF LARGE LANGUAGE MODELS (LLMS)

Large Language Models (LLMs) were used to aid in the writing and polishing of the manuscript. Specifically, we used an LLM to assist in refining the language, improving readability, and ensuring clarity in various sections of the paper. The model helped with tasks such as sentence rephrasing, grammar checking, and enhancing the overall flow of the text. It is important to note that the LLM was not involved in the ideation, research methodology, or experimental design. All research concepts, ideas, and analyses were developed and conducted by the authors. The contributions of the LLM were solely focused on improving the linguistic quality of the paper, with no involvement in the scientific content or data analysis. The authors take full responsibility for the content of the manuscript, including any text generated or polished by the LLM. We have ensured that the LLM-generated text adheres to ethical guidelines and does not contribute to plagiarism or scientific misconduct.

## A.2 MORE DISCUSSION AND DETAILS ABOUT *SFA* MODEL

The SFA module mainly consists of the Adaptive Filtering Selector (AFS) and the Information Concentrator Module (ICM). Our ICM structure is straightforward, consisting of only two linear layers and a frozen Llama-7B block. The first linear layer maps the visual features to the same dimension as Llama-7B, while the second linear layer maps it back to the original feature dimension. Both of these linear layers are trainable. After Filtering Attention, important visual features are aligned under the Llama module, rich in prior knowledge, achieving a focused effect. We further validate our approach by visualizing each layer in Fig. 5.

It can be observed that before entering the AFS module, the model focuses on non-target areas, resulting in a visual illusion. After passing through the AFS module, the model selects and filters the main target features, removing most of the redundant information. Then, the ICM module focuses on the key target features. Due to the absence of the SCL and NSM modules, some noise still remains in the final result.

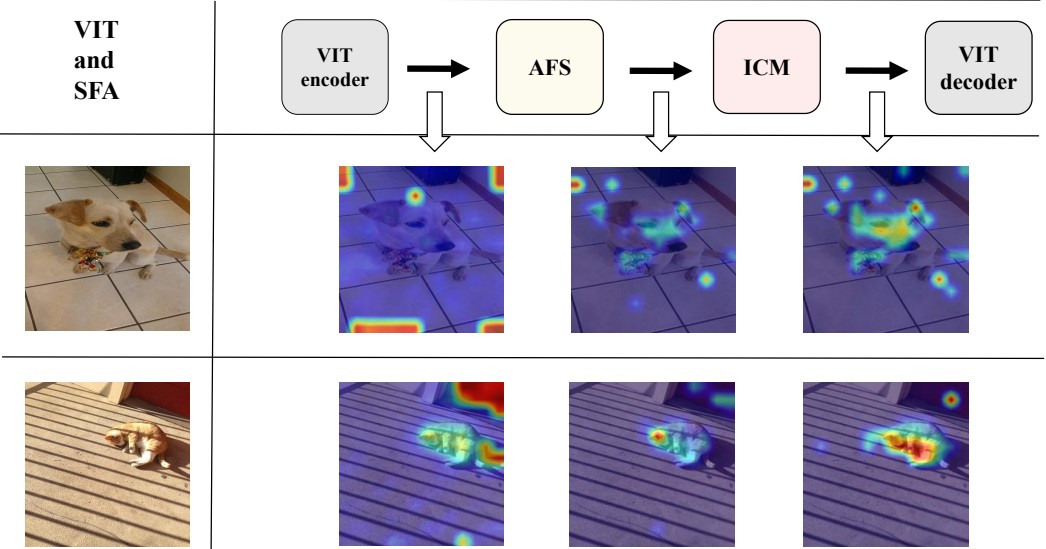

Figure 5: We conducted a visualization ablation experiment on the SFA module, where we obtained the attention map of each layer before entering the module using single Grad-CAM, in order to analyze the role played by the selection and focusing modules.

### A.3 MORE CROSS-DATASET EXPERIMENTAL RESULTS

We further conducted cross-dataset validation on more real and synthetic datasets. It is extremely challenging to span different quality content, and we achieved the best results among the existing mainstream comparison methods, further validating the effectiveness of our model.

Table 8: Performance comparison across different training and testing datasets

| Training | Testing | Re-IQA | Loda | Ours |
|----------|---------|--------|------|------|
| LIVE | CSIQ | 0.808 | 0.82 | 0.831 |
| - | TID2013 | 0.588 | 0.615 | 0.62 |
| CSIQ | LIVE | 0.929 | - | 0.933 |
| - | TID2013 | 0.575 | - | 0.608 |
| TID2013 | LIVE | 0.9 | 0.903 | 0.906 |
| - | CSIQ | 0.85 | 0.855 | 0.861 |
| Koniq | Tid2013 | 0.553 | 0.571 | 0.577 |

### A.4 THE COMPUTATIONAL COST DETAILS OF CSFIQA

The following four tables respectively show the training time (Tab. 9) and inference speed(Tab. 10) of CSFIQA compared with multiple SOTA methods on the Koniq dataset (Batch=64), the parameter composition of CSFIQA modules (Tab. 11), and the ablation experiments conducted on the filtering mechanism in the SFA module (Tab. 12).

Our initial intention was not to improve model speed, but to reduce the impact of information redundancy on model quality assessment. Based on your valuable suggestion, we tested our computational cost details of CSFIQA. Although our parameter count is nearly double that of Loda (SOTA), we have significantly reduced computational costs thanks to AFS (filtering mechanism)'s handling of information redundancy. When we removed the AFS module, the training time increased by nearly 3 times (Tab. 12). Notably, our large parameter count is mainly due to the frozen LLM module, with only 31M learnable parameters (Tab. 11).

Table 9: Training time comparison of different methods.

| Method | Per Epoch | Total Time |
|--------|-----------|------------|
| HyperIQA (Su et al., 2020a) | 928s | 11861s |
| LoDa (Xu et al., 2024) | 586s | 5798s |
| Ours | 270s | 2724s |

Table 10: Comparison of model parameters, MACs, and throughput.

| Method | Params | MACs | Throughput |
|--------|--------|------|------------|
| TReS (Golestaneh et al., 2022a) | 152.5M | 8.39G | 294(/s) |
| LoDa | 118.1M | 23.0G | 276(/s) |
| Ours | 233.2M | 45G | 515(/s) |

### A.5 FURTHER ABLATION STUDIES ON HYPERPARAMETERS

We further report ablation studies on the hyperparameters $\gamma_1$ and $\gamma_2$ in Eq.1, as well as the temperature hyperparameter $\tau$ in Eq.3. Our hyperparameters $\gamma_1$ and $\gamma_2$ are used for acquiring positive and negative samples, respectively. In Tab. 13, we report the performance of our model on LIVEC with different hyperparameter settings. The results indicate that our model performs best when $\gamma_1$ is set to 0.2 and $\gamma_2$ to 0.7. Similarly, we report the SRCC performance on LIVEC with different temperature $\tau$ in Tab. 14, with our model achieving optimal results when the temperature is set to 0.3.

Table 11: The detailed params of CSFIQA.

| params | total | trainable | frozen llm |
|--------|-------|-----------|------------|
| Ours   | 233M  | 31M       | 202M       |

Table 12: Ablation Study on Training Time for the AFS Module.

| Time | w/AFS | w/o AFS |
|------|-------|---------|
| Ours | 2724s | 8031s   |

## A.6 QUALITATIVE ANALYSIS

We use GradCAM to generate visual representations of the feature attention maps for the input images in our baseline model and CSFIQA, as shown in Fig. 6. Our proposed CSFIQA significantly outperforms the baseline because it better utilizes scale information to perceive image distortions. In contrast, the baseline is more prone to incorrectly focusing on non-distorted areas and exhibits "visual illusions". Our approach captures the complex relationships of different image quality regions, effectively extracting quality-aware features, highlighting our model's ability to capture accurate scale-quality variations and achieve more precise quality perception. This ability stems from our deliberate emphasis on regions with significant scale differences during training. Notably, the last two rows show the visualization results of our model at both large and small scales. Due to the presence of visual illusions, both visualizations focus on undistorted areas while ignoring the actual quality-critical regions. This aligns with our motivation. Additionally, the predicted quality scores further emphasize the superiority of our model compared to the baseline. In summary, the visualization results strongly validate the superiority of the proposed method.

## A.7 FAILURE CASE ANALYSIS

We present a failure case of CSIQA in Fig.7, which shows an underwater scene image. CSIQA captures incorrect distortion information because in underwater scenes, distortions are typically global, such as blur, haze, and other distortion types that cover the entire image. However, the CSIQA method emphasizes visual illusions caused by quality information differences across multiple scales. Obviously, in underwater scenes, image patches of different sizes have essentially similar distortion characteristics. This cross-scale distortion consistency does not align with the fundamental assumptions of CSIQA's multi-scale approach. CSFIQA relies on detecting quality variations between image patches at different resolutions, which typically exist in natural and synthetic images, but underwater images exhibit consistent overall degradation patterns regardless of patch size. Therefore, the cross-scale comparison mechanism cannot identify meaningful quality differences, leading to inaccurate quality assessment.

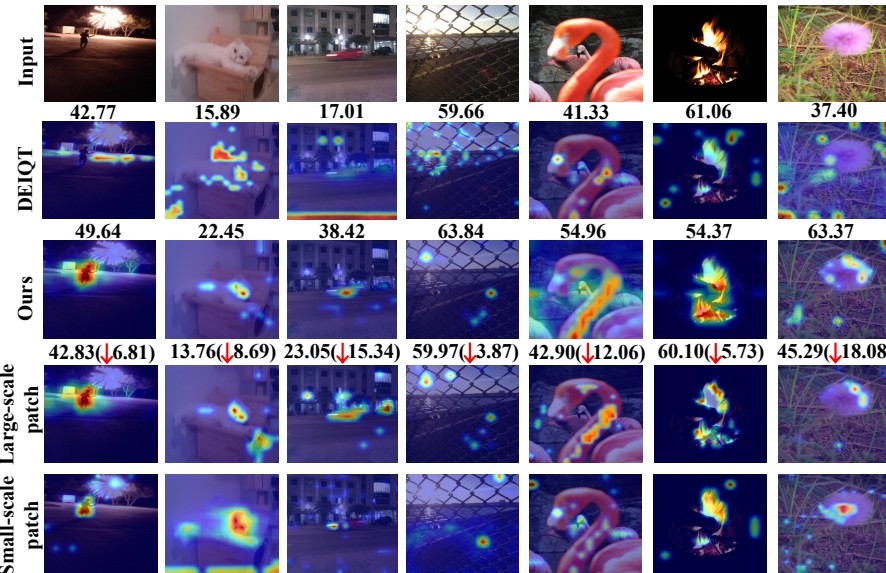

Figure 6: Activation maps of baseline model and CSFIQA using Grad-CAM (Selvaraju et al., 2017) on authentic dataset LIVEC. The scores below the first row of images represent the images' ground truth mos. In contrast, our model focuses more on the distorted regions of the image, resulting in our image quality predictions being closer to the true values. Rows 1 to 3 show the input image, CAM from baseline, and CAM from CSFIQA, respectively. Rows 4 and 5 further present the visualization results of large and small scale in CSFIQA features.

Table 13: The performance of CSFIQA in terms of SRCC on the LIVE dataset with different values of $[\gamma_1, \gamma_2]$.

| $[\gamma_1, \gamma_2]$ | [0.1,0.5] | [0.1,0.8] | [0.2,0.7] | [0.2,0.8] | [0.3,0.9] |
|---|---|---|---|---|---|
| CSFIQA | 0.897 | 0.903 | 0.905 | 0.900 | 0.883 |

Table 14: The performance of CSFIQA in terms of SRCC on the LIVE dataset with different values of $\tau$. The best performance is achieved when $\tau$ is set to 0.3. Therefore, we set the hyperparameter $\tau$ to 0.3.

| $\tau$ | 0.1 | 0.2 | 0.3 | 0.4 | 0.5 |
|---|---|---|---|---|---|
| CSFIQA | 0.902 | 0.901 | 0.905 | 0.904 | 0.900 |

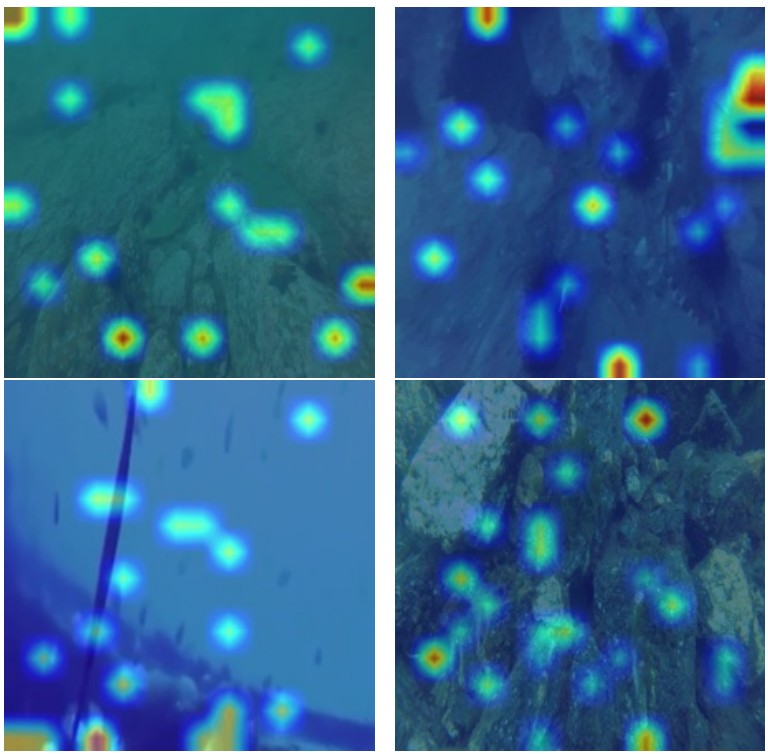

Figure 7: Failure examples of CSFIQA in underwater scenes.

