# OpenReview forum: "Scale Contrastive Learning with Selective Attentions for Blind Image Quality Assessment"
_ICLR.cc/2026/Conference — ICLR 2026 Conference Withdrawn Submission_

### Official Review · Reviewer_uaWk · 2025-10-24

**Soundness:** 2
**Presentation:** 2
**Contribution:** 2
**Rating:** 2
**Confidence:** 5

**Summary:**

This paper proposes CSFIQA, a multi-scale blind image quality assessment (BIQA) framework that integrates two main innovations:
(1) a Selective Focus Attention (SFA) mechanism designed to filter redundant cross-scale information and concentrate on quality-relevant cues, and
(2) a Scale Contrastive Learning (SCL) strategy, combined with an adaptive Noise Sample Matching (NSM) module, to model scale-dependent quality differences and mitigate the so-called “visual illusion” problem in existing BIQA models.

The authors evaluate CSFIQA on seven public IQA datasets (LIVE, CSIQ, TID2013, LIVEC, KonIQ-10K, LIVEFB, SPAQ) and report superior SRCC/PLCC performance over 16 baselines (e.g., DEIQT, LoDa, QFM-IQM). Ablation and visualization analyses suggest that both SCL and SFA modules contribute to the improvement.

**Strengths:**

1) The paper evaluates across seven standard datasets (synthetic and authentic distortions), performs cross-dataset tests, and includes ablations on hyperparameters (λ, [α, β], τ). This extensive coverage indicates careful empirical effort.

2) The reported +8.8% SRCC on LIVEFB and solid results on KonIQ-10k and LIVEC suggest that the proposed modules capture some useful cross-scale cues, validating the importance of scale-aware modeling.

3) The inclusion of detailed module-wise ablations (SCL vs. SFA vs. NSM) and computational statistics in the appendix allows partial reproducibility and insight into design sensitivity.

**Weaknesses:**

1) The combination of contrastive learning, multi-scale encoding, and selective attention follows directly from existing BIQA pipelines (CONTRIQUE, Re-IQA, MUSIQ, LoDa). The work lacks a novel theoretical formulation or perceptual model explaining why the specific design choices improve human alignment.

2) Terms like “visual illusion” and “information dilution” are used as rhetorical devices rather than measured phenomena. There is no statistical evidence that cross-scale fusion actually causes these effects.

3) The “Information Concentrator Module” uses a frozen LLaMA-7B, but the rationale and mechanism are opaque. No ablation quantifies its contribution; its inclusion increases parameters from 118M (LoDa) to 233M with negligible improvement.

4) The experimental setup may favor CSFIQA: the pre-trained CrossViT backbone and different patch sizes might lead to stronger features than prior models. Fair comparison requires matching architecture capacity or retraining baselines under equivalent compute.

**Questions:**

1) Have authors measured perceptual inconsistency across scales using subjective ratings or MOS re-annotation to confirm this hypothesis?

2) What motivates using LLaMA-7B for attention concentration instead of a lightweight vision-only transformer or MLP? How does it contribute beyond dimensional mapping?

3) How were the thresholds (γ₁=0.2, γ₂=0.7) selected? Could the model benefit from adaptive thresholding or temperature scaling tuned per dataset?

4) Given the large model (233M parameters, 45 G MACs), how does the performance compare if the frozen LLM is removed or replaced with a smaller encoder?

5) Were all baselines trained and evaluated under identical data splits, augmentation, and epochs? Some methods (e.g., LoDa, DEIQT) use different pre-training strategies; please clarify to ensure fair comparison.

6) How would CSFIQA perform on AIGC-IQA datasets or other-field IQA, where cross-scale distortion consistency differs (as you briefly note in Sec. A.7)?

7) Could this approach extend to temporal quality assessment or multimodal IQA (video, 3D, or generated content)? If so, how would scale contrastive principles transfer?

8)  How does SCL differ in principle from prior quality-aware contrastive objectives that already consider multi-scale or distortion-level variations? Does SFA introduce a new attention mechanism, or does it merely re-parameterize the top-k masking already seen in transformer pruning or efficient attention literature?

---

### Official Review · Reviewer_SE3v · 2025-10-30

**Soundness:** 3
**Presentation:** 3
**Contribution:** 2
**Rating:** 2
**Confidence:** 4

**Summary:**

The paper proposes CSFIQA, a new blind image quality assessment (BIQA) framework designed to better align with human multi-scale visual perception. The authors argue that existing multi-scale BIQA models fail because they naively fuse features across scales, creating “visual perception conflicts” and redundant information that weakens quality-sensitive cues.

**Strengths:**

1. The performance is good.
2. The figures are well drawn.

**Weaknesses:**

1. The paper should better illustrate how the attention mechanism identifies redundant vs. informative cross-scale features.
2. The contrastive learning process needs clearer formulation. How to define positive or negative pairs.
3. Multi-scale models can be heavy. The paper should report: parameters, FLOPs, inference speed, relative to baselines. This is essential for real-time/embedded use cases.

**Questions:**

1. What is intra-scale contrastive learning and inter-scale contrastive learning?
2. What is the function of cross-attention.
3. Why using cross-attention between small patches and larger patches rather than feature fusion. This may increase computational cost and may not useful.

---

### Official Review · Reviewer_tKTC · 2025-10-31

**Soundness:** 2
**Presentation:** 2
**Contribution:** 2
**Rating:** 4
**Confidence:** 4

**Summary:**

The proposed CSFIQA presents a conceptually interesting and technically promising framework that models scale-dependent perception in Blind Image Quality Assessment (BIQA). The paper introduces two key innovations: Scale Contrastive Learning (SCL) and Selective Focus Attention (SFA), aiming to mitigate “visual illusions” and “information dilution” across scales. The idea of explicitly learning quality relationships between scales is novel and timely.

**Strengths:**

+ Recognizes and formalizes the long-neglected problem of scale-dependent perceptual variation, offering a new perspective for BIQA.
+ Provides GradCAM visualizations illustrating attention improvements over baselines.
+ Covers eight datasets (synthetic and authentic), with ablations on all modules and hyperparameters.

**Weaknesses:**

- The conceptual definition of "visual illusion" and its mathematical mapping to the contrastive loss is vague. It is unclear how the “illusion” manifests quantitatively and why contrastive learning inherently solves it.
- The SCL formulation (Eq.1–3) lacks theoretical justification. Why should MOS-based pairwise distances define positive/negative relations? Are these thresholds robust across datasets?
- Whether the cross-dataset results (Tab. 2) are trained on synthetic → authentic or vice versa affects generalization claims.
- The contrastive learning novelty overlaps with Re-IQA and CONTRIQUE. The paper should clarify how scale contrastive learning differs from existing sample-level contrastive strategies
-  The paper claims improved speed due to information filtering (Appendix A.4), but the parameter count is large (233M). The frozen LLM dominates computation.

**Questions:**

- The paper lacks a clear description of the validation set or model selection protocol. While the datasets are split into training and testing sets (80/20), it is unclear whether a separate validation set was used for hyperparameter tuning, early stopping, or model selection. This omission raises potential concerns about data leakage or overfitting, especially since the model integrates several modules (SCL, NSM, SFA) with multiple tunable parameters.
- Typos: Line 164/245: “pacth” → “patch.”; Appendix A.6: “CSIQA” should be “CSFIQA.”

---

### Official Review · Reviewer_Kj3d · 2025-10-31

**Soundness:** 3
**Presentation:** 3
**Contribution:** 2
**Rating:** 4
**Confidence:** 3

**Summary:**

This paper introduces the Contrast-Constrained Scale-Focused Image Quality Assessment (CSFIQA) framework to overcome two key limitations in multi-scale BIQA: "visual illusions" and "information dilution," which compromise human perception alignment. The core contribution lies in the Scale Contrastive Learning (SCL) strategy, augmented by Noise Sample Matching (NSM), which explicitly models quality discrepancies across and within different visual scales. Complementing this, a Selective Focus Attention (SFA) mechanism filters redundant cross-scale information, thereby enhancing sensitivity to subtle, quality-critical features. Evaluations conducted across seven benchmark datasets confirm that CSFIQA achieves superior performance over state-of-the-art methods.

**Strengths:**

Strengths
- The paper clearly identifies specific challenges of "visual illusions" and "information dilution" that plague traditional multi-scale BIQA approaches.
- The Scale Contrastive Learning (SCL) framework, utilizing MOS similarity to select positive and negative pairs, and the Noise Sample Matching (NSM) mechanism, which targets regional quality variations, represent a novel and effective strategy for mitigating scale-dependent perceptual distortions.
- The proposed CSFIQA achieves SOTA results. The model demonstrates robust generalization capabilities, achieving the best performance across various cross-dataset validation experiments.

**Weaknesses:**

Weaknesses
- One of the main motivations, "visual illusions" (the phenomenon where quality perception changes with scale), is not directly reflected in the classification component of SCL. Instead, SCL merely uses MOS to classify positive and negative pairs. Although NSM is designed to address this issue, the SCL component itself appears to adopt a method with the same limitations (regarding visual illusions) that the paper's motivation criticizes in existing approaches.
- NSM relies on the strong assumption that ViT feature similarity equates to quality similarity, rather than the semantic similarity of patches. This assumption may be invalid if the "information dilution" problem persists. Therefore, more rigorous experiments are needed to validate that the model's ViT features primarily emphasize quality-related components to rationalize NSM.
- While the paper proposes a loss function and architecture that overcome limitations within the specific field of IQA, its theoretical contribution from a general machine learning perspective appears weak. The work is limited to the narrow domain of IQA and, as a result, may not have a significant impact on the general ICLR community.
- Given the paper's focus on scale-dependent quality variations, it would be beneficial to add experiments. These experiments should independently analyze how this phenomenon manifests for different distortion types and demonstrate how the proposed method addresses each.
- Despite the AFS improving training time, the overall computational demand, indicated by a high MACs value (45G) compared to competitors like TReS (8.39G), suggests the model may be computationally heavy during inference.

Minor
- It should be mentioned that Eq. 3 is identical to the InfoNCE loss.
- Incomplete descriptions of notations F+ and F- in Eq. 3
- The text in most of the main figures is too small.

**Questions:**

Please check the major weaknesses.

---

### Note · Authors · 2025-11-23

I have read and agree with the venue's withdrawal policy on behalf of myself and my co-authors.